# Genetic and Evolutionary Analysis of Canine Coronavirus in Guangxi Province, China, for 2021–2024

**DOI:** 10.3390/vetsci11100456

**Published:** 2024-09-26

**Authors:** Kaichuang Shi, Yandi Shi, Yuwen Shi, Yi Pan, Shuping Feng, Zhuo Feng, Yanwen Yin, Yang Tang, Zongqiang Li, Feng Long

**Affiliations:** 1School of Basic Medical Sciences, Youjiang Medical University for Nationalities, Baise 533000, China; panyiyi2004@163.com; 2College of Animal Science and Technology, Guangxi University, Nanning 530005, China; shiyandi123@126.com (Y.S.); shiyuwen2@126.com (Y.S.); fz2187941887@163.com (Z.F.); 18020762192@163.com (Y.T.); zqingli20@gxu.edu.cn (Z.L.); 3Guangxi Center for Animal Disease Control and Prevention, Nanning 530001, China; fsp166@163.com (S.F.); yanwen0349@126.com (Y.Y.)

**Keywords:** canine coronavirus (CCoV), S gene, M gene, N gene, phylogenetic analysis, genetic diversity

## Abstract

**Simple Summary:**

Canine coronavirus (CCoV) is an important gastrointestinal pathogen in dogs. It is endemic worldwide and poses a significant risk to canine health. Here, the prevalence and genetic characteristics of CCoV in pet dogs in Guangxi Province, China, were examined. Sixty-five CCoV-positive samples were selected for amplifying and sequencing S, M, and N genes to analyze the similarity, recombination, and evolution of CCoV strains. The results indicated that the positivity rate of CCoV was 8.43% and showed a close correlation with age and season in pet dog infections. The strains obtained in Guangxi Province belonged to the CCoV-IIa and CCoV-IIb subtypes, and recombination and mutations were found in Guangxi’s CCoV strains. These provided insights into the genetic characteristics of CCoV in Guangxi Province, China.

**Abstract:**

Canine coronavirus (CCoV) is an important gastrointestinal pathogen that causes serious harm to pet dogs worldwide. Here, 1791 clinical samples were collected from pet dogs in different pet hospitals in Guangxi Province, southern China, for the 2021–2024 period and detected for CCoV by a multiplex RT-qPCR. The results showed that 8.43% (151/1791) of samples were positive for CCoV. Sixty-five positive samples were selected to amplify, sequence, and analyze S, M, and N genes. A sequence comparison revealed that the nucleotide and amino acid similarities of the S, M, and N genes were 94.86% and 94.62%, 96.85% and 97.80%, and 96.85% and 97.80%, respectively. Phylogenetic analysis indicated that 65 CCoV strains obtained in this study belonged to the CCoV-II genotype, of which 56 CCoV strains belonged to the CCoV-IIa subtype and 9 CCoV strains belonged to the CCoV-IIb subtype. A potential recombination event analysis of S gene sequences indicated that two CCoV strains, i.e., GXBSHM0328-34 and GXYLAC0318-35, have recombination signals. A Bayesian analysis indicated that the evolutionary rates of the S, M, and N genes were 1.791 × 10^−3^, 6.529 × 10^−4^, and 4.775 × 10^−4^ substitutions/site/year, respectively. The population size grew slowly before 1980 and then began to shrink slowly; it then shrank rapidly in 2005 and expanded sharply in 2020, leveling off thereafter. These results indicated the CCoV strains prevalent in Guangxi Province, southern China, showed a high level of genetic diversity and maintained continuous variation among clinical epidemic strains.

## 1. Introduction

Canine coronavirus (CCoV) is a member of the genus *Alphacoronavirus* in the *Coronaviridae* family, with a single-stranded, positive-sense RNA genome [1]. CCoV infection was first reported in 1971 when a strain of coronavirus (1–71 strain) was isolated from dogs suffering from acute enteritis in a German military dog unit [2]. Since then, many reports in different countries have confirmed CCoV as a major gastrointestinal pathogen in dogs. CCoV is considered a source of viral infection that seriously endangers canine farming and wildlife conservation. With a high prevalence of infection and a low mortality rate, it is an important virus that causes varying degrees of gastroenteritis in dogs [3,4]. Dogs of different ages, breeds, and genders are infected with CCoV, with higher morbidity and mortality in puppies [5]. CCoV often co-infects with other gastrointestinal viruses, for example, canine parvovirus (CPV), canine distemper virus (CDV), and canine rotavirus (CRV) [6,7,8,9,10].

The CCoV genome is approximately 28–32 kb and is composed of the following genes: ORF1ab, S, ORF3ac, E, M, N, and ORF7ab [11]. The globally predominant CCoV strains have two genotypes, which are designated CCoV-I and CCoV-II [12], and CCoV-II is further classified into subtypes CCoV-IIa and CCoV-IIb [13,14,15]. A new variant of CCoV-II (CCoV-IIv) has been identified in recent studies but has not yet been officially recognized [16].

Coronaviruses have four genera: α, β, γ, and δ. CCoV is classified into the α-coronavirus genus, which differs significantly from the other three genera in its S gene. The nucleotide similarity values for CCoV with canine respiratory coronavirus (CRCoV) and 2019-nCoV of the β-coronavirus genus are 42.0–50.6% and 34.4–46%, respectively. The nucleotide similarity values for CCoV with feline coronavirus (FCoV) and porcine transmissible gastroenteritis virus (TGEV), which also belong to α-coronavirus, are 42.0–51.0% and 55.2–92.6%, respectively. As shown in the phylogenetic tree based on different coronaviruses (Appendix A), Figure 1, CCoV is closely related to FCoV and TGEV but has low similarity with other coronaviruses.

Pet dogs have close relationships with humans. Some human infections with novel CCoV have been reported [17,18,19]. A novel canine/feline recombinant α-coronavirus (CCoV-HuPn-2018) was isolated from a patient with pneumonia in Malaysia [17], and a CCoV strain (HuCCoV_Z19) was isolated from a patient in Haiti which shared 99.4% nucleic acid identity with CCoV-HuPn-2018 [18], suggesting the zoonotic potential of CCoV [19]. The potential for these strains to cause serious diseases in humans cannot be ignored; thus, CCoV requires more attention.

In this study, clinical samples from pet dogs were collected in 23 pet hospitals in 8 regions in Guangxi Province, southern China, and tested for the presence of CCoV using an RT-qPCR previously developed in our laboratory. The date of collection, location, prevalence, and correlation between gender and season of the samples were analyzed, and the genetic and evolutionary characteristics of the CCoV strains were examined to provide basic data for CCoV prevention and control strategies.

## 2. Materials and Methods

### 2.1. Clinical Sample Collection and Test

Between December 2021 and March 2024, in 23 pet hospitals in 8 regions of Guangxi Province, southern China, a total of 1791 nasal and anal swabs and fecal samples were collected from 1791 pet dogs that were asymptomatic or demonstrated enteritis signs of diarrhea and/or vomiting. All samples were transported to our laboratory under ≤4 °C conditions and diluted in phosphate-buffered saline (PBS, pH 7.2). They were centrifuged at 12,000 rpm for 2 min, and 200 μL supernatants were used to extract viral RNA using a TaKaRa MiniBEST Nucleic Acid Extraction Kit (Dalian, China). All 1791 clinical samples were detected for CCoV using our previously developed quadruplex RT-qPCR [6]. Briefly, the following specific primers and probes were used: CCoV(M)-F/R: GGTGGTATGAACATCGACAATT/TTAGATTTTACATAGTAAGCCCATCC; CCoV(M)-P: FAM-CGTAATGGTTGCATTACCTAGCAGGACCAT-BHQ1. The following amplification procedure was used: 42 °C for 5 min, followed by 95 °C for 10 s, and 40 cycles at 95 °C for 5 s, followed by 56 °C for 34 s. A Ct value ≤ 36 indicated a positive sample.

Of the 151 CCoV-positive clinical samples tested via quadruplex RT-qPCR, 65 samples were selected based on their sampling regions, sampling dates, and Ct values. The extracted total nucleic acids were reverse-transcribed to cDNA according to the former report [6] and stored at −80 °C before use.

### 2.2. PCR Amplification of Samples

#### 2.2.1. Primer Design

The reference sequences of CCoV from the NCBI GenBank (https://www.ncbi.nlm.nih.gov/nucleotide/, accessed on 10 September 2021) were compared, and specific primers targeting the S, M, and N genes of CCoV were designed (Table 1). To facilitate sequence splicing, the amplified fragments of CCoV S gene were overlapped. All primers were synthesized by TaKaRa Co., Ltd. (Dalian, China).

#### 2.2.2. Sequencing of Target Fragments

The S, M, and N gene fragments of the 65 selected positive samples were amplified using PCR. The PCR reaction system, with a total volume of 25 μL, was as follows: 12.5 μL of TaKaRa Premix Taq™ (Dalian, China), 0.4 μL of forward/reverse primer each, 9.2 μL of distilled water, and 2.5 μL of viral cDNA. The amplification procedures were as follows: 94 °C for 3 min; 35 cycles of 94 °C for 30 s, 56 °C–58 °C (CCoV-M/S4: 58 °C; CCoV-N: 57 °C; CCoV-S1/S2/S3: 57 °C; CCoV-S5: 56 °C) for 30 s, 72 °C for 40 s–75 s (CCoV-M/S4: 60 s; CCoV-N: 75 s; CCoV-S1/S2/S3: 72 s; CCoV-S5: 40 s); and 72 °C for 10 min.

The PCR products were purified using a TaKaRa MiniBEST Nucleic Acid Extraction Kit (Dalian, China), ligated into TaKaRa pMD18-T Vector (Dalian, China), and transformed into TaKaRa *E. coli* DH5α competent cells (Dalian, China). The positive clones were cultured in an LB liquid medium containing ampicillin at 37 °C for 20–24 h; plasmids were extracted from the bacterial solution and sequenced by Guangzhou IGE Biotechnology Co., Ltd. (Guangzhou, China). The sequencing results were spliced using SeqMan (DNASTAR 7.1) software (https://www.dnastar.com/software/, accessed on 10 September 2021), and a nucleotide/amino acid similarity matrix was generated using BioAider software V1.727 (https://github.com/ZhijianZhou01/BioAider/releases, accessed on 10 September 2021) [20].

### 2.3. CCoV Prevalence

#### 2.3.1. CCoV Time Distribution Analysis in China

Canine coronavirus, CCoV in China, canine diarrhea, and canine virus epidemics were used as keywords, and 6270 publications related to the prevalence of CCoV infections in China in the last ten years from 2014 to 2023 were identified from NCBI PubMed (https://pubmed.ncbi.nlm.nih.gov/, accessed on 14 March 2024), CNKI (https://www.cnki.net/, accessed on 14 March 2024), CSTJ Database (https://www.cqvap.com, accessed on 14 March 2024), and Wanfang Database (https://www.wanfangdata.com.cn, accessed on 14 March 2024). In addition, the positive cases of CCoV (146/1536, 9.50%) identified in this study from 2021 to 2023 were included in the analysis.

#### 2.3.2. Investigation of CCoV in Guangxi Province

To examine the prevalence of CCoV in Guangxi Province, a total of 1791 samples collected from dogs in 23 pet hospitals in 8 regions of Guangxi between December 2021 and March 2024 were analyzed based on sex, age, health status, sampling date, and sampling location.

### 2.4. Phylogenetic and Recombination Analyses

A phylogenetic tree was generated via the maximum likelihood method, using 1000 bootstrap replications in MEGA X 10.2.6 software (https://www.megasoftware.net/archived_version_active_download, accessed on 14 March 2024); then, the Interactive Tree of Life (iTOL) (https://itol.embl.de/, accessed on 14 March 2024) was used to optimize the phylogenetic tree. The S, M, and N genes of CCoV, FCoV, and TGEV strains sampled within and outside of China were collected from the nucleotide database of the NCBI GenBank. Seven algorithms were selected for recombination analysis using Recombination Detection Program (RDP4) V4.0 software (http://www.bioinf.manchester.ac.uk/recombination/programs, accessed on 14 March 2024), including RDP, Chimaera, BootScan, 35 eq, GENECONV, MaxChi, and SiScan, and the potential recombination results of the sequences were consistent when at least five algorithms were supported. Potential recombination sequences were validated using SimPlot 3.5.1 software (https://github.com/Stephane-S/Simplot_PlusPlus, accessed on 14 March 2024). CCoV is closely related to TGEV and FCoV, demonstrating a high genetic correlation [17,18,21,22]; therefore, FCoV and TGEV were also involved in the phylogenetic analysis in this study.

### 2.5. Estimation of the Evolutionary Rates

The S, M, and N gene sequences of the 65 obtained CCoV strains and the reference strains from the NCBI GenBank were compared using MEGA X 10.2.6 software; then, the temporal signals of the S, M, and N gene sequences were evaluated using TempEst software (version 1.5.3) (http://beast.community/tempest, accessed on 14 March 2024). Suitable arithmetic models were developed using the Model Finder option in PhyloSuite software (version 1.2.2) (http://phylosuite.jushengwu.com/, accessed on 14 March 2024). The uncorrelated relaxed clock and Bayesian skyline models were finally chosen as the best models for the S, M, and N genes, in addition to alternative GTR+F+I+G4, TN93+F+I+G4, and TN93+F+G4 models for the S, M, and N genes, respectively. They were run in BEAST version 1.10.4 software (http://beast.community/, accessed on 14 March 2024) with a chain length of 200 million. Data convergence (ESS > 200) was performed in Tracer software (version 1.6) (https://github.com/beast-dev/tracer/releases/latest, accessed on 14 March 2024).

## 3. Results

### 3.1. Analysis of Temporal Distribution of CCoV in China

An analysis of CCoV in China over the past 10 years shows that the prevalence of CCoV infection gradually increased from 2014 to 2016, reaching its highest at 52.52% in 2016, and decreased from 2017 to 2020. During the 2021–2023 period, the prevalence of CCoV infection slightly increased and remained stable with little difference. The average infection rate was 26% (Figure 2).

### 3.2. Test Results of Clinical Samples

The 1791 clinical samples were detected using a previously developed quadruplex RT-qPCR [6]; 151 (151/1791, 8.43%) samples were positive for CCoV. The positivity rates of clinical samples for 2021–2024 were 11.11% (2/18), 8.45% (54/639), 9.88% (90/911), and 2.22% (5/223), respectively, and the positivity rates in 8 regions of Guangxi Province ranged from 0% (0/11) in Beihai to 21.65% (84/338) in Baise (Figure 3). The 65 CCoV-positive samples originating from different regions and dates were selected for amplification and sequencing, and 65 S, 65 M, and 65 N gene sequences were obtained and uploaded to the NCBI GenBank database under the accession numbers PP583073–PP583137 for the S gene, PP583138–PP583202 for the M gene, and PP583203–PP583267 for the N gene (Appendix A).

### 3.3. Prevalence Analysis of CCoV in Guangxi Province

The prevalence of CCoV in the 1791 dogs in this study was analyzed for statistical significance by sex, age, clinical signs, sampling date, and sampling location using the Chi-square test (Table 2). The results showed that the prevalence rates of CCoV infection in dogs under 6 months of age, 6–24 months of age, and over 2 years of age were 12.93%, 5.14%, and 4.27%, respectively, indicating that puppies were more susceptible to CCoV (*p* < 0.05). CCoV was also tested in subclinical dogs in addition to diseased animals, with an infection rate of 2.66% (*p* < 0.05). The statistical results for CCoV infection rates in different seasons showed that CCoV occurred in all seasons, with infection rates ranging from 5.05% to 13.69%, and spring and autumn were seasons in which dogs were more susceptible seasons to CCoV (*p* < 0.05). No statistical difference in the prevalence of CCoV infection by gender was found). As for sampling locations/sites, the Baise region had the most serious rate of CCoV infection, with an infection rate of 21.65%, and the Nanning, Yulin, and Guilin regions had similar infection rates of 5.29%, 6.72%, and 5.75%, respectively, while the Liuzhou and Qinzhou regions had even lower infection rates of 3.76% and 0.53%, respectively (Figure 3).

### 3.4. Sequence Similarity Analysis

The S, M, and N gene sequences of the CCoV strains were downloaded from the NCBI GenBank (Appendix A). The average nucleotide similarities of the S, M, and N genes of the obtained 65 CCoV strains from Guangxi Province were 94.86% (77.36–100%), 96.85% (92.68–100%), and 96.85% (85.01–99.91%), respectively, and the average amino acid similarities were 94.62% (71.50–100%), 97.80% (93.52–100%), and 97.80% (92.36–100%), respectively. The 65 CCoV strains from Guangxi Province had 86.04% (49.89–100%), 93.22% (79.00–100%), and 95.31% (76.28–100%) nucleotide similarities for the S, M, and N genes, respectively, with the reference CCoV strains obtained domestically and abroad. In addition, they had 85.66% (36.28–100%), 94.59% (78.72–100%), and 95.85% (73.00–100%) amino acid similarities for the S, M, and N genes, respectively, with the reference CCoV strains obtained domestically and abroad.

### 3.5. Phylogenetic Trees Based on CCoV S, M, and N Genes

#### 3.5.1. Phylogenetic Trees of S Gene Sequences

A total of 208 S gene sequences were analyzed in this study, including 65 CCoV sequences from Guangxi Province and 102 CCoV, 33 FCoV, and 8 TGEV reference sequences obtained domestically and abroad that were downloaded from the NCBI GenBank (Appendix A). They were subjected to sequence comparison using MEGA X software, and GTR+G+I was confirmed to be the best model. Then, a phylogenetic tree was generated via the ML method, using 1000 bootstrap tests. As shown in Figure 4, all CCoV strains were classified into genotype I (CCoV-I) and genotype II (CCoV-II), and the CCoV-II strains were further divided into the subtypes CCoV-IIa and CCoV-IIb. All 65 CCoV strains from Guangxi Province (marked by black circles ●) were classified as CCoV-II. Among these, 56 strains belonged to CCoV-IIa, forming a large branch with other Chinese strains, while the remaining 9 strains belonged to CCoV-IIb. These strains were most closely related to the Vietnamese strain CCoV/dog/HCM47/2015 (GenBank accession No. LC290907) and the American strain CCoV/7/2020/AUS (GenBank accession No. MW383487).

#### 3.5.2. Phylogenetic Trees of M Gene Sequences

A total of 217 M gene sequences were analyzed in this study, including 65 CCoV sequences from Guangxi Province and 118 CCoV, 27 FCoV, and 7 TGEV reference sequences obtained domestically and abroad that were downloaded from the NCBI GenBank (Appendix A). They were subjected to sequence comparison using MEGA X software, and TN93+G+I was confirmed to be the best model. Then, the phylogenetic tree was generated via the ML method with 1000 bootstrap tests. As shown in Figure 5, all CCoV strains were classified into CCoV-I and CCoV-II. The 65 CCoV strains from Guangxi Province (marked with the black circle ●) were classified as CCoV-II. Among these, some strains belonged to the same branch as the strains from the USA, the UK, and Haiti, but most strains belonged to the same branch as the Chinese strains.

#### 3.5.3. Phylogenetic Trees of N Gene Sequences

A total of 204 N gene sequences were analyzed in this study, including 65 CCoV sequences from Guangxi Province and 104 CCoV, 28 FCoV, and 7 TGEV reference sequences from within and outside China which were downloaded from the NCBI GenBank (Appendix A). They were subjected to sequence comparison using MEGA X software, and TN93+G was confirmed to be the best model. Then, the phylogenetic tree was generated using the ML method with 1000 bootstrap tests. As shown in Figure 6, all CCoV strains were classified into CCoV-I and CCoV-II. The 65 CCoV strains from Guangxi Province (marked with black circles ●) were classified as CCoV-II. Among these, most strains were closely related to the Chinese strains, while the other strains were closely related to the US and UK strains.

### 3.6. Bayesian Temporal Dynamics Analysis

The time-history origins of CCoV were analyzed based on the S gene. The average evolutionary rate of the CCoV-II S gene was estimated to be 1.791 × 10^−3^ substitutions/site/year (s/s/y) (95% HPD: 1.013 × 10^−3^–2.458 × 10^−3^ s/s/y). Since all the obtained Guangxi CCoV strains belonged to CCoV-II, the obtained strains and downloaded reference strains (Appendix A) were used for maximum clade credibility (MCC) tree analysis according to CCoV-II S gene sequences. The results indicated that all the CCoV strains were divided into CCoV-IIa and CCoV-IIb subtypes, most of the obtained strains from Guangxi Province were classified as CCoV-IIa, and the other strains were classified as CCoV-IIb (Figure 7). The Bayesian skyline showed that the CCoV population size had grown slowly before 1980 (Figure 8). After that, it began to shrink slowly. However, in 2005, it began to shrink rapidly before expanding quickly in 2020. Although it has since begun to level off, this is still higher than the overall size of the population.

### 3.7. Analysis of Amino Acid Variations in CCoV S Gene

The S gene amino acid sequences of the 65 CCoV strains obtained in this study and the classical reference strains were compared to analyze the S gene variants of the prevalent CCoV strains in Guangxi Province. Taking CCoV-IIa classical strains 1–71 and CB/05 as reference strains, the 56 CCoV-IIa strains showed amino acid-induced mutations at eight sites, i.e., N^204^→T^204^, E^255^/D^255^→N^255^, V^266^→I^266^, L^711^→I^711^, S^721^→A^721^, G^897^→S^897^, V^1280^→A^1280^, and L^1284^→C^1284^ (Figure 9A,B). Compared with the classical human-originated strains CCoV-HuPn-2018 and HuCCoV-Z19, there was one amino acid mutation in nine strains of CCoV-IIb, i.e., T^806^→I^806^ (Figure 9D). In addition, compared with these two human-derived CCoV strains, the obtained strains in this study and the reference strains had nine additional amino acid mutations, i.e., L^321^→I^321^, I^397^→T^397^, D^408^→E^408^, H^640^→Y^640^, V^719^-I^719^, V^1119^→A^1119^, Q^1248^→E^1248^, C^1253^→M^1253^, and E^1317^→V^1317^ (Figure 9C,D).

### 3.8. Analysis of Recombination in S Gene

The recombination of the S gene sequences of the CCoV strains from Guangxi Province was analyzed using RDP4 and SimPlot software. The results revealed that recombination signals were detected in the GXBSHM0328-34 strain (Accession No. PP583106) and the GXYLAC0318-35 strain (Accession No. PP583107) (Figure 10). The major parent of GXBSHM0328-34 is the JS2103 strain (CCoV-IIa, GenBank accession No. OM055788), with 98.9% similarity, and the minor parent is the GH4-2 strain (CCoV-IIa, GenBank accession No. OM950729) with 96.8% similarity. There are potential breakpoints in the 859 nt–1110 nt and 1766 nt–2009 nt regions. The major parent of the GXYLAC0318-35 strain is GXBSHM0213-31 (CCoV-IIa, GenBank accession No. PP583103), with 99.7% similarity, and the minor parent is the C21110201 strain (CCoV-IIa, GenBank accession No. OQ351915), with 98.1% similarity. There are potential breakpoints in the 2254 nt–2650 nt region.

### 3.9. Evolutionary Rates Analysis

The evolutionary rates of the S, M, and N genes of the CCoV strains from Guangxi Province were estimated via BEAST software (v 1.10.4) and Bayesian clustering analysis. The results indicated that the evolutionary rates of the S, M, and N genes were 1.791 × 10^−3^, 6.529 × 10^−4^, and 4.775 × 10^−4^ s/s/y, respectively (Table 3).

## 4. Discussion

In recent years, the health of companion animals has come under scrutiny, and their infection with a wide range of viruses has attracted great attention. Dogs infected with CCoV show gastrointestinal symptoms such as diarrhea and vomiting, and severely infected puppies may even die. Data on CCoV infections in China collected from databases such as NCBI PubMed and CNKI show that CCoV continuously infects dogs in China. Thus, more attention should be paid to the trend in CCoV persisting in dog populations in different regions of China. In this study, 1791 samples of clinically sick and asymptomatic/subclinical dogs from 23 pet hospitals in 8 regions of Guangxi Province showed a positivity rate of 8.43% for CCoV, indicating that CCoV is widely prevalent in Guangxi Province. Compared with other reports, the positivity rate of CCoV in Guangxi was lower than that in Japan (50.5%, 55/109) [23], Italy (28.21%, 11/39) [9], and the pooled prevalence of CCoV infection in eight provinces (Gansu, Heilongjiang, Jilin, Beijing, Tianjin, Shandong, Jiangsu, and Henan) of China (33%, 3604/10,609) [22]. The overall prevalence in other regions was as follows: Heilongjiang (28.36%, 57/201) [24], Chengdu (27.06%, 59/218) [25], Beijing (26.02%, 64/246) [26], and five other provinces (Guangdong, Zhejiang, Heilongjiang, Jiangsu, and Anhui) in China (23.94%, 51/213) [21]. In Brazil, it was 12.00% (30/250) [27], which is slightly higher than in sick dogs in India (5.89%, 28/475) [10]. Factors such as different environments, regions, ages, and health conditions can lead to different infection rates. Epidemiological observations of 1791 dogs from Guangxi province in this study showed that puppies were more susceptible to CCoV than adult dogs, and dogs were infected with CCoV more in spring and autumn, which was consistent with the results of other studies in China [3,26,28]. Notably, CCoV was tested in clinical samples taken from subclinical dogs with a positivity rate of 2.41%, and infected but subclinical dogs may act as important carriers of pathogens during epidemic outbreaks, which could pose a serious public health problem [29,30]. Therefore, the prevalence and variants of CCoV strains in Guangxi Province must be investigated further, and extensive molecular analyses of circulating strains are essential to addressing vaccination failure. Studies have also reported that canine–cat–pig recombinant α-coronavirus has been detected in humans. This virus has been related to human respiratory disease [17,18,19]. These data highlight the zoonotic potential of coronaviruses and their threat to public health, indicating that coronavirus surveillance needs to be strengthened.

The average nucleotide and amino acid similarities of the S, M, and N genes of the 65 CCoV strains from Guangxi Province were 94.86% and 94.62%, 96.85% and 97.80%, and 96.85% and 97.80%, respectively, suggesting there existed higher variation in the S gene than in the M and N genes. The phylogenetic trees (Figure 4, Figure 5 and Figure 6) revealed that 65 strains from Guangxi Province were classified as CCoV-IIa (56 strains) and CCoV-IIb (9 strains) based on the S gene and were distributed in CCoV-II based on the M and N genes. These results indicated that CCoV-IIa was the predominant subtype prevalent in Guangxi Province, while CCoV-IIb was also prevalent but less frequent, a result consistent with the prevalence previously reported in China [3,22]. In addition, the obtained strains showed mutations in different regions in the S gene (Figure 9), and 2 of 65 strains showed recombinants in the S gene (Figure 10). The different nucleotide and amino acid similarities, subtype distribution, S gene mutations, and recombinants indicate that the CCoV strains prevalent in Guangxi Province have high genetic diversity.

CCoV S gene amino acid sequence analysis revealed that the 56 Guangxi strains of CCoV-IIa had eight mutations compared with the classical strains 1–71 and CB/05, and the other 9 Guangxi strains of CCoV-IIb had one mutation compared with classical human-derived strains CCoV-HuPn-2018 and HuCCoV_Z19 [17,18,19]. It was also observed that the canine-derived reference CCoV strain and Guangxi CCoV-IIb strains had nine additional amino acid mutations compared to the two human-derived CCoV-IIb CCoV strains. The trans-species transmission and zoonotic potential of CCoV might be attributed to viral variations and recombinants [17,18,19]. Therefore, the mutations in these amino acid sites need to be continuously observed and estimated.

The average evolutionary rates of the S, M, and N genes were 1.791 × 10^−3^, 6.529 × 10^−4^, and 4.775 × 10^−4^ s/s/y, respectively, indicating that the S gene had higher variation rates than the M and N genes. These rates were similar to the previously reported results [31]. Bayesian analysis of the MCC tree indicated that the population size began to shrink rapidly in 2005 before expanding dramatically in 2020. Although it began to level off thereafter, it was still higher than the overall population magnitude. Several studies have shown that recombination can generate a large number of variant genotypes and facilitate the generation of new strains [13,14,15,16]. The presence of CCoV-IIb strains due to the recombination of the S genes of CCoV-II and TGEV suggests possible canine–porcine intraspecies transmission [13,32]. In contrast, the double homologous recombination of FCoV-I and CCoV resulted in FCoV-II [33,34]. These results suggest that the recombination of CCoV cannot be ignored. The S gene sequences of the 65 CCoV strains from Guangxi Province were analyzed for potential recombination events, and recombination signals were found in GXBSHM0328-34 and GXYLAC0318-35 strains, increasing the genetic diversity of CCoV strains. It is worth noting that if significant genetic variation occurs in the virus, it may weaken or defeat the efficacy of vaccine protection, increasing the prevalence of CCoV.

## 5. Conclusions

This epidemiological survey indicated that CCoV had a positivity rate of 8.43% in pet dogs from Guangxi Province in southern China during the 2021–2024 period; this positivity rate was closely related to age and season. A phylogenetic analysis based on CCoV’s S, M, and N gene sequences revealed that CCoV-IIa was the predominant genotype prevalent in Guangxi Province; CCoV-IIb was also prevalent but was less frequent. The CCoV population grew slowly before 1980 and began to shrink slowly afterward. It began to shrink rapidly in 2005 and then expanded sharply in 2020 before leveling off. The CCoV strains prevalent in Guangxi Province in 2021–2024 showed high diversity and rapid evolution, and more effective measures should be taken to prevent and control the prevalence and spread of CCoV in pet dog populations.

## Figures and Tables

**Figure 1 vetsci-11-00456-f001:**
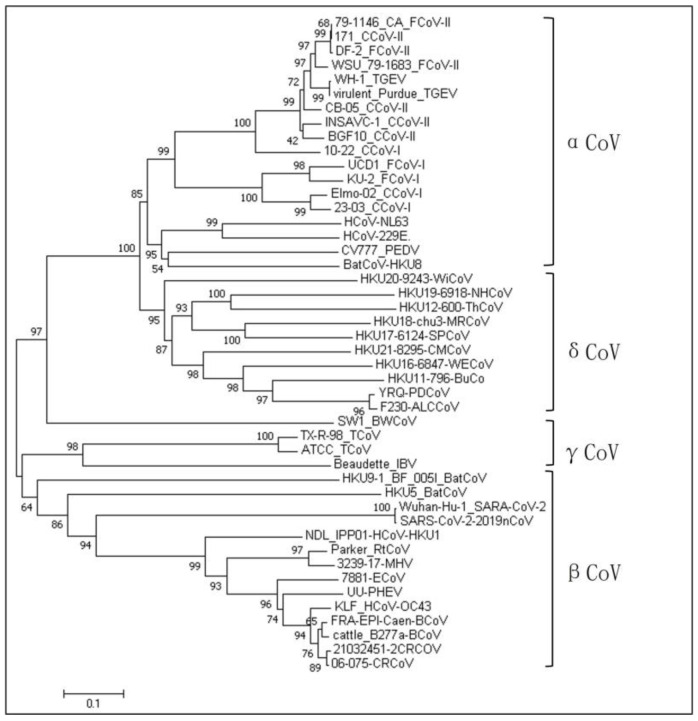
Genetic evolutionary relationships of different species of coronaviruses.

**Figure 2 vetsci-11-00456-f002:**
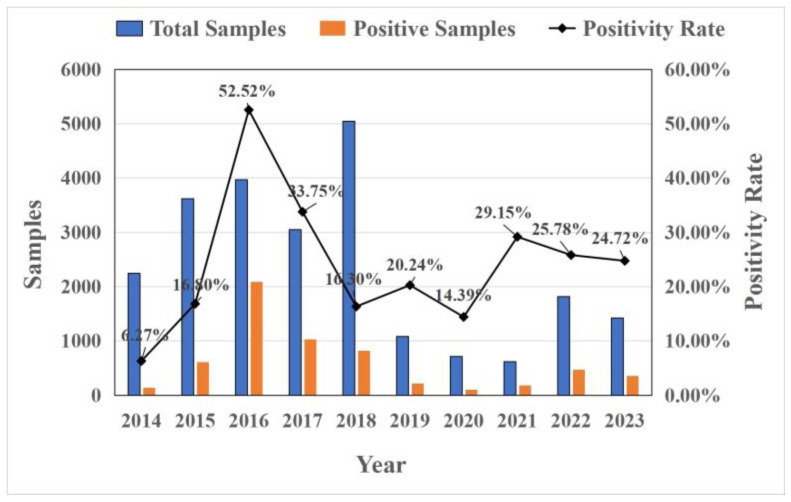
The prevalence of CCoV infection in China in 2014–2023.

**Figure 3 vetsci-11-00456-f003:**
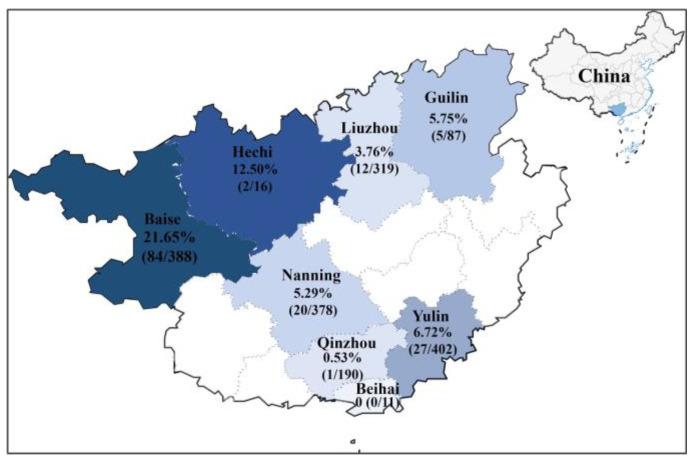
Distribution of CCoV in Guangxi Province, China.

**Figure 4 vetsci-11-00456-f004:**
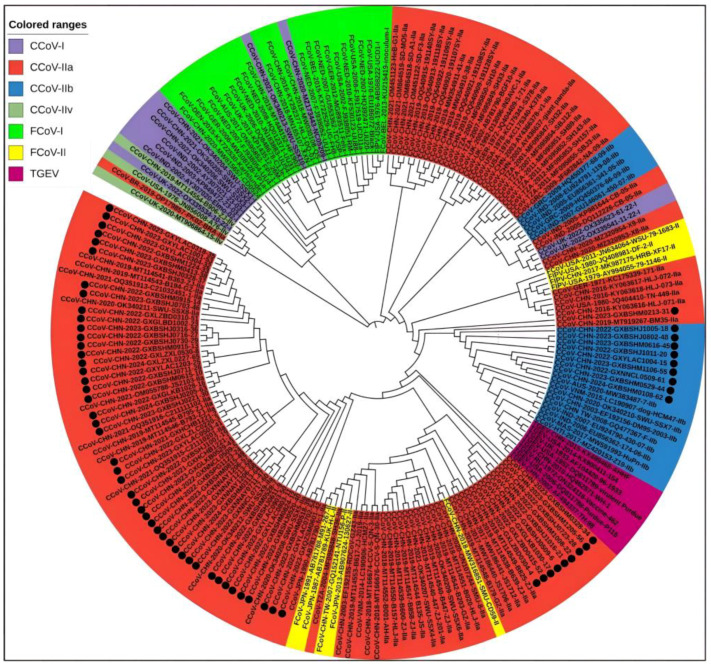
The CCoV S gene phylogenetic tree. The 65 CCoV strains obtained in this study were marked with black circles (●).

**Figure 5 vetsci-11-00456-f005:**
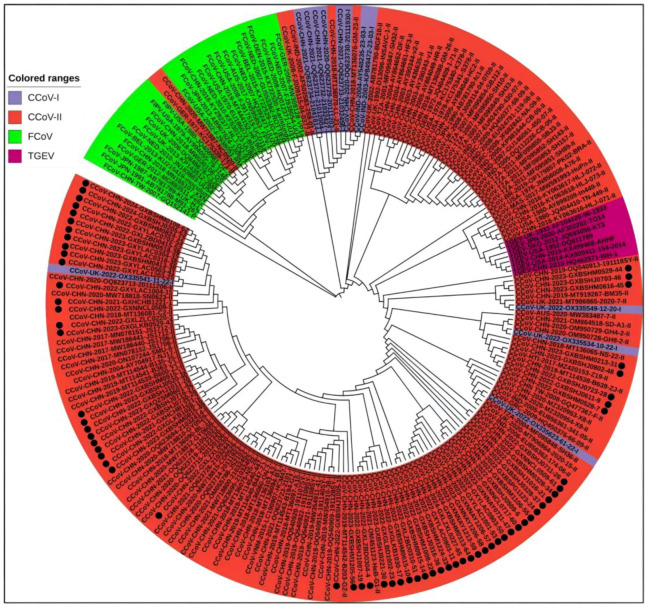
The CCoV M gene phylogenetic tree. The obtained 65 CCoV strains from Guangxi Province in this study were marked with black circles (●).

**Figure 6 vetsci-11-00456-f006:**
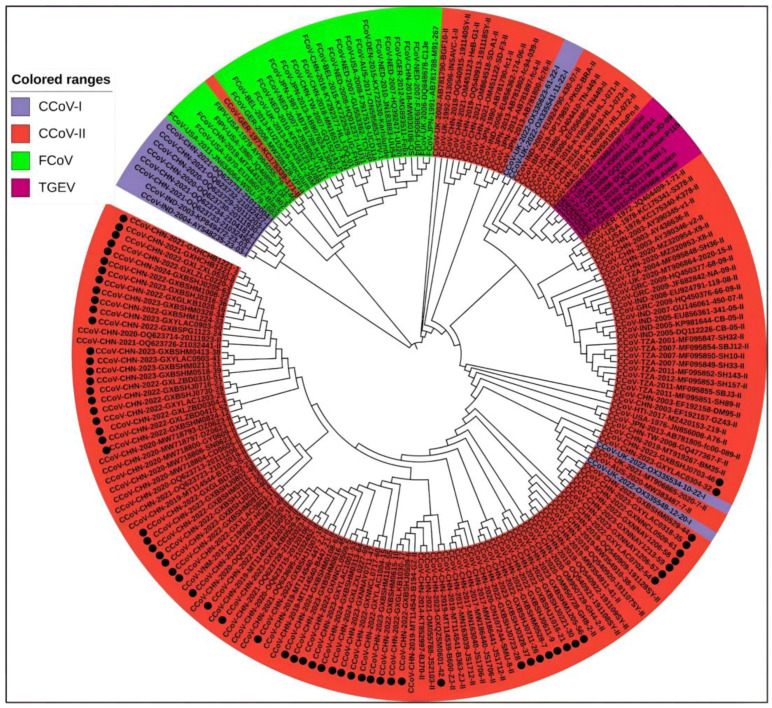
The CCoV N gene phylogenetic tree. The obtained 65 CCoV strains from Guangxi Province in this study are marked with black circles (●).

**Figure 7 vetsci-11-00456-f007:**
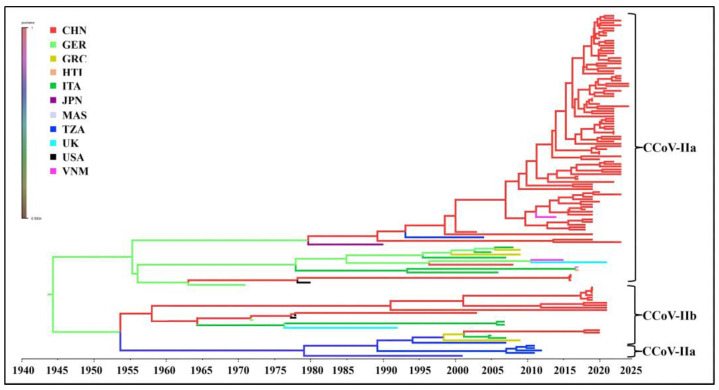
Maximum clade credibility (MCC) tree of CCoV-II S gene sequences.

**Figure 8 vetsci-11-00456-f008:**
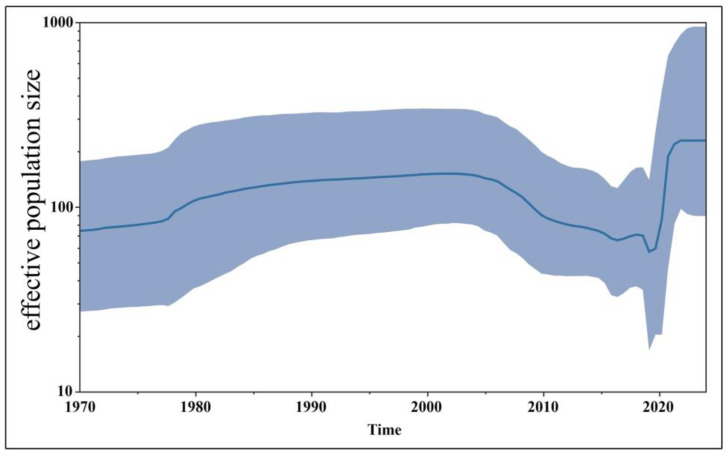
The demographic history of the CCoV-II strains. The blue line indicates the median estimate of the effective population size, and the shaded regions indicate the corresponding 95% confidence interval.

**Figure 9 vetsci-11-00456-f009:**
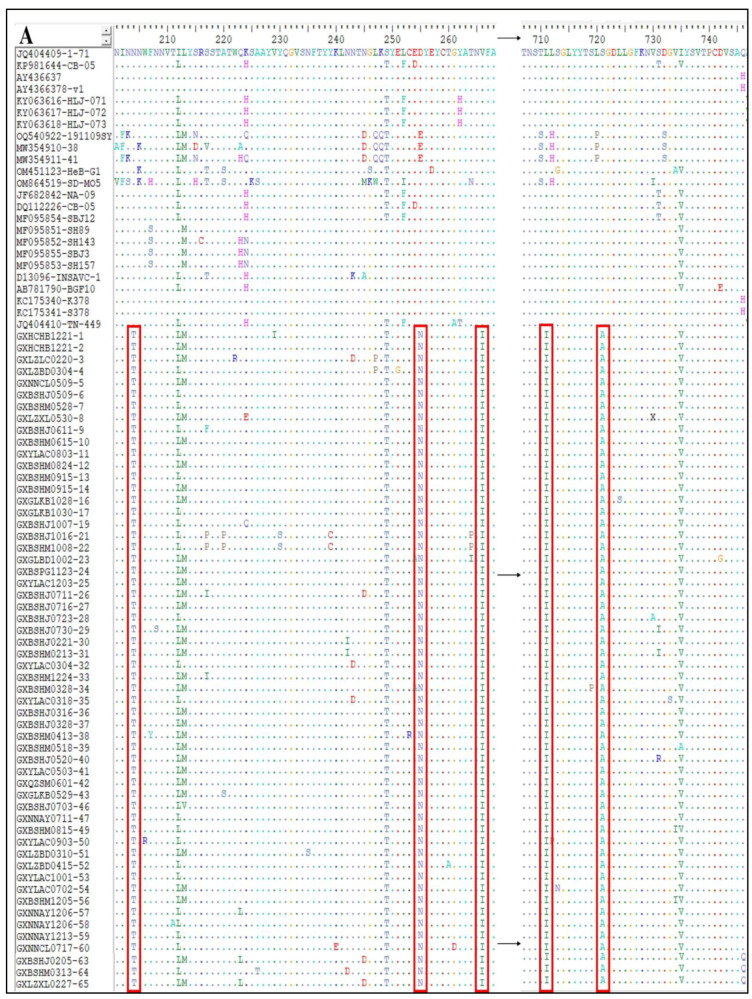
Mutations of amino acids encoded by CCoV S gene. (**A**,**B**) CCoV-IIa strain; (**C**,**D**) CCoV-IIb strain.

**Figure 10 vetsci-11-00456-f010:**
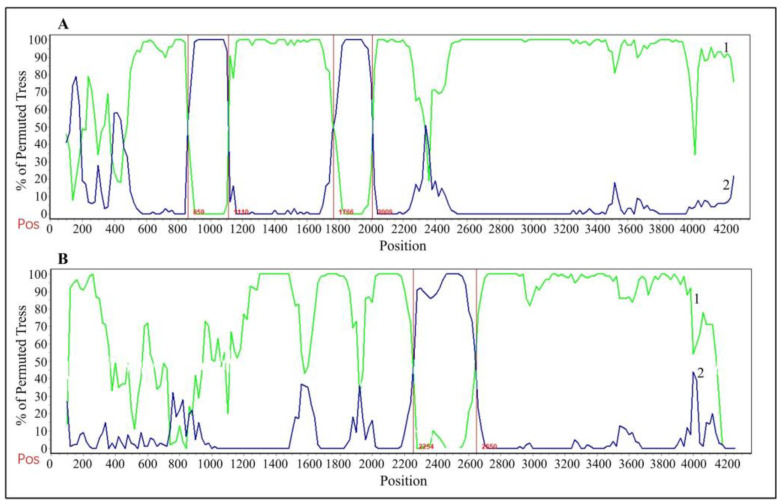
Recombination analysis of CCoV S gene sequences. (**A**) GXBSHM0328-34 strain; (**B**) GXYLAC0318-35 strain. 1: minor parent; 2: major parent.

**Table 1 vetsci-11-00456-t001:** The primers for PCR amplification.

Gene	Primer	Sequence (5′→3′)	Tm (°C)	Product/bp
M	CCoV-M-F	CCTGATGAAGCACTCCTTGTT	58.7	911
	CCoV-M-R	GGCCACGAGAATTGGAAAGA	54.5	
N	CCoV-N-F	TCAACAGACGCAAGAACTGATA	55.9	1225
	CCoV-N-R	TCGTTTAGTTCGTCACCTCATC	57.4	
S	CCoV-S1-F	ATGGTCGTTGGATTACTAAG	50.9	1180
	CCoV-S1-R	GCACCCATGCCAGATTGTAC	58.7	
	CCoV-S2-F	GTTAATTGYTTRTGGCCAGT	54.5	1154
	CCoV-S2-R	CAGGAGTCTAAGTGTARCACTGAC	57.2	
	CCoV-S3-F	TGGTGCCAACTGYAAGTTTGATG	58.5	1190
	CCoV-S3-R	GTGCACCTAATGTTATACCACCTGC	60.7	
	CCoV-S4-F	TACCTGGTGTGGCTAATGATGAC	58.5	901
	CCoV-S4-R	GGTATAATACTAGGCAAGTCAATTAC	54.5	
	CCoV-S5-F	GACTCCCAGAACTATGTATCAGCC	59.9	638
	CCoV-S5-R	CCATACAAGACCTGTAATGACTC	57.0	

**Table 2 vetsci-11-00456-t002:** The CCoV infection in 1791 dogs from Guangxi Province.

Variables		Number of Dogs (*n* = 1791)	CCoV-Positive Dogs (*n* = 151)	CCoV-Negative Dogs (*n* = 1640)	*p* Value
Sex	Female	1052	96 (9.13%)	956 (90.87%)	>0.05
	Male	739	55 (7.44%)	684 (92.56%)	
Age	<6 months	776	107 (13.79%)	674 (86.86%)	<0.05
	6–24 months	560	27 (4.82%)	533 (95.18%)	
	>2 years	455	17 (3.74%)	438 (92.26%)	
Clinical status	Asymptomatic	831	20 (2.41%)	811 (97.59%)	<0.05
	Sick	960	131 (13.65%)	829 (86.35%)	
Season	Spring (Mar–May)	431	59 (13.69%)	372 (86.31%)	<0.05
	Summer (Jun–Aug)	515	26 (5.05%)	489 (94.95%)	
	Autumn (Sep–Nov)	473	47 (9.94%)	426 (90.06%)	
	Winter (Dec–Feb)	372	19 (5.11%)	353 (94.89%)	

**Table 3 vetsci-11-00456-t003:** The evolutionary rates of CCoV S, M, and N genes.

Gene	Evolutionary Rate (s/s/y)	95% HPD (s/s/y)
S	1.791 × 10^−3^	1.013 × 10^−3^–2.458 × 10^−3^
M	6.529 × 10^−4^	4.817 × 10^−4^–8.2772 × 10^−4^
N	4.775 × 10^−4^	3.607 × 10^−4^–5.8693 × 10^−4^

## Data Availability

Data are contained within the article and Appendix A.

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
