# Peer review of "Genetic and Evolutionary Analysis of Canine Coronavirus in Guangxi Province, China, for 2021–2024"

_vetsci, 2024, doi:10.3390/vetsci11100456_

Round 1

Reviewer 1 Report

Comments and Suggestions for Authors

The manuscript entitled "Genetic and Evolutionary Analysis of Canine Coronavirus in Guangxi Province of China during 2021-2024” was evaluated. Researchers reported that CCoV had a positivity rate of 8.43% in pet dogs from Guangxi province in southern China during 2021-2024, which were closely related to age and season, and The CCoV prevalent strains in Guangxi province showed high diversity and rapid evolution. This manuscript helps to understand of the epidemiology of CCoV in China. The paper is in the scope of the journal and may be published.

Negative aspects

line 301-303:  This inference is not appropriate in ‘results section’, and could be discussed in ‘discussion section’,

line 364-376: these information has been documented in ‘results section’.

line 380-388: reference should be added in this part.

Author Response

The Cover Letter (vetsci-3189883)

September 20, 2024

Dear editor,

Our manuscript has been revised carefully according to the suggestions of editor and reviewers. The details are as follows.

Reviewer #1

Comments and Suggestions for Authors

The manuscript entitled "Genetic and Evolutionary Analysis of Canine Coronavirus in Guangxi Province of China during 2021-2024” was evaluated. Researchers reported that CCoV had a positivity rate of 8.43% in pet dogs from Guangxi province in southern China during 2021-2024, which were closely related to age and season, and The CCoV prevalent strains in Guangxi province showed high diversity and rapid evolution. This manuscript helps to understand of the epidemiology of CCoV in China. The paper is in the scope of the journal and may be published.

Negative aspects

  1. line 301-303: This inference is not appropriate in ‘results section’, and could be discussed in ‘discussion section’.

Response: We agree to the reviewer’s suggestion. This sentence has been deleted, and related contents has been added in discussion section. Please see Lines 340-341, and 404-406 in the revised manuscript.

  1. line 364-376: This information has been documented in ‘results section’.

Response: We agree to the reviewer’s suggestion. This sentence has been re-written. The contents similar to those in the result section have been deleted. Please see Lines 407-422 in the revised manuscript.

  1. line 380-388: Reference should be added in this part.

Response: We agree to the reviewer’s suggestion. References have been added in this part. Please see Lines 434, and 438 in the revised manuscript.

Reviewer #2

Comments and Suggestions for Authors

Title: Genetic and Evolutionary Analysis of Canine Coronavirus in Guangxi Province of China during 2021-2024. The main objective was to collect clinical samples of pet dogs from 23 pet hospitals in 8 regions of Guangxi to detect the presence of CCoV. The date of collection, location, prevalence, and correlation between gender and season of the samples were analysed, and the genetic and evolutionary characteristics of CCoV strains prevalent in Guangxi province were analysed. This manuscript is interesting, but needs to be improved the professional English editing to be published.

  1. The use of professional vocabulary is not accurate.

Response: We agree to the reviewer’s suggestion. The manuscript has been finished English editing by MDPI (Author Services ID: english-85167). Please see the revised manuscript.

  1. Line 92-93 a quadruplex RT-qPCR established in our laboratory, please briefly describe the test steps.

Response: We agree to the reviewer’s suggestion. The test steps of the quadruplex RT-qPCR has been described briefly. Please see Lines101-108 in the revised manuscript.

  1. 2.3.2 Why the two time points of December 2021 and March 2024 were selected for sample collection?

Response: Thanks for the reviewer’s suggestion. This time is related to the implementation period of the funded project provided for this study.

  1. What is the statistical method used in Table 2 ?

Response: The Chi-square test was used in Table 2. Please see Lines 220 in the revised manuscript.

  1. Please improve the quality of the Figure.

Response: We agree to the reviewer’s suggestion. The figures 4,5, and 6 have been revised to increase their quality. Please see Figure 4, 5, and 6 in the revised manuscript.

  1. Please add the discussion of the results of Figure 4, Figure 5 and Figure 6.

Response: We agree to the reviewer’s suggestion. The results of Figure 4, Figure 5 and Figure 6 have been discussed in the discussion section. Please see Lines 410-413 in the revised manuscript.

Comments on the Quality of English Language

  1. This manuscript is interesting, but needs to be improved the professional English editing to be published.

Response: We agree to the reviewer’s suggestion. The manuscript has been finished English editing by MDPI (Author Services ID: english-85167). Please see the revised manuscript.

Reviewer #3

Comments and Suggestions for Authors

  1. Sample size was not statistically determined and sampling was not randomized.

Response: Thanks for the reviewer’s suggestion. The clinical samples were collected from 23 pet hospitals in eight regions of Guangxi Province during 2021-2024. The sample size was depended on the number of cases of sick dogs coming to the hospitals for treatment. The samples were collected randomly.

  1. What was the basis for selection of 65 out of 151 CCoV for sequencing of S, M, N genes?

Response: Thanks for the reviewer’s suggestion. Of the 151 CCoV-positive clinical samples, 65 positive samples were selected based on the sampling regions, sampling dates, and Ct value of the samples. Please see Lines 109-111 in the revised manuscript.

  1. In line 27, note that 65 strains were not obtained in the study, but those selected for further analyses.

Response: In this study, 65 S, 65 M, 65 N gene sequences were obtained from clinical samples, and were used for genetic and evolutionary analysis. Please see Lines 25-27 in the revised manuscript.

  1. Line 94: Specify the number of CCoV positive clinical samples.

Response: We agree to the reviewer’s suggestion. The number of CCoV positive clinical samples has been added. Please see Line 109 in the revised manuscript.

  1. Line 158: Is not referring to sequences of 65 CCoV strains obtained in the study, but sequences of three genes, namely S, M, and N.

Response: We agree to the reviewer’s suggestion. The S, M, and N gene sequences of 65 CCoV strains obtained in the study were used for genetic and evolutionary analysis. This information has been added in the revised manuscript. Please see Line 178 in the revised manuscript.

  1. Lines 171-177 and 181-191, authors are mixing up results presentation with discussion, which should be separated.

Response: We agree to the reviewer’s suggestion. Two paragraphs have been revised according to the reviewer’s suggestion. Please see Lines 193-200, and 203-214 in the revised manuscript.

  1. Line 360: Should be referring to Canine-cat-pig recombinant alpha-coronavirus.

Response: We agree to the reviewer’s suggestion. The word “recombinant” has been added in the revised manuscript. Please see Line 402 in the revised manuscript.

 Comments on the Quality of English Language

  1. Several grammatical, tense, plural and phraseology challenges with the write-up.

Response: We agree to the reviewer’s suggestion. The manuscript has been finished English editing by MDPI (Author Services ID: english-85167). Please see the revised manuscript.

In addition, the manuscript has been revised carefully according to the editor’s suggestion to decrease the total similarity rate lower than 30% and the single similarity rate lower than 5%. Please see the revised manuscript.

Best regards,

Kaichuang Shi

Reviewer 2 Report

Comments and Suggestions for Authors

Title: Genetic and Evolutionary Analysis of Canine Coronavirus in Guangxi Province of China during 2021-2024. The main objective was to collect clinical samples of pet dogs from 23 pet hospitals in 8 regions of Guangxi to detect the presence of CCoV. The date of collection, loca-tion, prevalence, and correlation between gender and season of the samples were analysed, and the genetic and evolutionary characteristics of CCoV strains prevalent in Guangxi province were analysed. This manuscript is interesting, but needs to be improved the professional English editing to be published.

1、The use of professional vocabulary is not accurate.

2、Line 92-93 a quadru- 92 plex RT-qPCR established in our laboratory, please briefly describe the test steps.

3、2.3.2 Why the two time points of December 2021 and March 2024 were selected for sample collection?

4、What is the statistical method used in Table 2 ?

5、Please improve the quality of the Figure.

Please add the discussion of the results of Figure 4, Figure 5 and Figure 6.

Comments on the Quality of English Language

This manuscript is interesting, but needs to be improved the professional English editing to be published.

Author Response

(The authors gave the same response as above.)

Reviewer 3 Report

Comments and Suggestions for Authors

1. Sample size was not statistically determined and sampling was not randomized.

2. What was the basis for selection of 65 out of 151 CCoV for sequencing of S, M, N genes?

3. In line 27, note that 65 strains were not obtained in the study, but those selected for further analyses.

4. Line 94: Specify the number of CCoV positive clinical samples

5. Line 158: Is not referring to sequences of 65 CCoV strains obtained in the study, but sequences of three genes, namely S, M, and N.

6. Lines 171-177 and 181-191, authors are mixing up results presentation with discussion, which should be separated.

7. Line 360: Should be referring to Canine-cat-pig recombinant alpha-coronavirus

Comments on the Quality of English Language

Several grammatical, tense, plural and phraseology challenges with the write-up.

Author Response

(The authors gave the same response as above.)
